# Depressed Myocardial Energetic Efficiency Increases Risk of Incident Heart Failure: The Strong Heart Study

**DOI:** 10.3390/jcm8071044

**Published:** 2019-07-17

**Authors:** Maria-Angela Losi, Raffaele Izzo, Costantino Mancusi, Wenyu Wang, Mary J. Roman, Elisa T. Lee, Barbara V. Howard, Richard B. Devereux, Giovanni de Simone

**Affiliations:** 1Hypertension Research Center, University Federico II of Naples, I-80131 Naples, Italy; 2Department of Advanced Biomedical Sciences, University Federico II of Naples, I-80131 Naples, Italy; 3College of Public Health, University of Oklahoma Health Sciences Center, Oklahoma City, OK 73104, USA; 4Department of Medicine, Weill Cornell Medical College, New York, NY 10065, USA; 5Center for American Indian Health Research, University of Oklahoma Health Sciences Center, Oklahoma City, OK 73126, USA; 6Medstar Health Research Institute, and Georgetown-Howard Universities Center for Translational Sciences, Washington, DC 20057, USA

**Keywords:** left ventricular hypertrophy, heart failure with preserved ejection fraction, population study, stroke volume, heart rate, echocardiography

## Abstract

An estimation of myocardial mechano-energetic efficiency (MEE) per unit of left ventricular (LV) mass (MEEi) can significantly predict composite cardiovascular (CV) events in treated hypertensive patients with normal ejection fraction (EF), after adjustment for LV hypertrophy (LVH). We have tested whether MEEi predicts incident heart failure (HF), after adjustment for LVH, in the population-based cohort of a “Strong Heart Study” (SHS) with normal EF. We included 1912 SHS participants (age 59 ± 8 years; 64% women) with preserved EF (≥50%) and without prevalent CV disease. MEE was estimated as the ratio of stroke work to the “double product” of heart rate times systolic blood pressure. MEEi was calculated as MEE/LV mass, and analyzed in quartiles. During a follow-up study of 9.2 ± 2.3 years, 126 participants developed HF (7%). HF was preceded by acute myocardial infarction (AMI) in 94 participants. A Kaplan-Meier plot, in quartiles of MEEi, demonstrated significant differences, substantially due to the deviation of the lowest quartile (*p* < 0.0001). Using AMI as a competing risk event, sequential models of Cox regression for incident HF (including significant confounders), demonstrated that low MEEi predicted incident HF not due to AMI (*p* = 0.026), after adjustment for significant effect of age, LVH, prolonged LV relaxation, diabetes, and smoking habits with negligible effects for sex, hypertension, antihypertensive therapy, obesity, and hyperlipemia. Low LV mechano-energetic efficiency per unit of LVM, is a predictor of incident, non-AMI related, HF in subjects with initially normal EF.

## 1. Introduction

Heart failure (HF) is predominantly a disease of the elderly, with nearly 50% of patients having preserved (p) left ventricular (LV) ejection fraction (EF) [1]. Although mechanisms for HFpEF remain incompletely understood, diastolic dysfunction, because it underlies myocardial hypertrophy and fibrosis, is thought to play a dominant role [2]. However, diastolic dysfunction also occurs in systolic HF (HFrEF) and is also common in elderly hypertensive individuals without HF [3]. Thus, abnormalities other than diastolic function are likely to be involved in HFpEF, especially in the presence of LV hypertrophy (LVH) [4].

There is evidence that pressure-overload LVH preserves EF as a measure of LV systolic function at the chamber level, even when contractility is reduced at the level of cardiomyocytes [5]. However, when paralleling the magnitude of LV chamber dimensions, even at normal values of LV systolic chamber function, important differences can occur in the magnitude of stroke volume (SV, i.e. LV pump performance), heart rate (HR), and blood pressure (BP) [6]. As a consequence, at a given EF, hemodynamic workload might differ substantially [6] and, in fact, SV is more predictive of incident HF than EF [7].

A new parameter of LV performance has been recently proposed as a surrogate measure of myocardial mechano-energetic efficiency (MEE), which is the ratio between produced external systolic work (stroke work, SW) and an estimate of myocardial oxygen consumption (MVO2) [8]. MEE per unit of LV mass (MEEi) has been demonstrated to predict composite adverse cardiovascular (CV) events in treated hypertensive patients after adjustment for LVH [9].

One unexplored issue is whether MEEi can also help explain incidence of HF, after adjustment for LVH, in the presence of initial normal EF in population studies. Accordingly, this analysis has been designed to assess whether MEEi can improve the identification of phenotypes at high risk of incident HF in members of the “Strong Heart Study” (SHS) cohort who were free of prevalent cardiovascular (CV) disease and initially had normal EF.

## 2. Methods

### 2.1. Participants

We analyzed data from the SHS, a population-based cohort study of CV risk factors and disease in American Indians. Detailed descriptions of the study design and methods have been previously reported [10]. At time of enrollment, a total of 4549 American Indian men and women, aged 45 to 74 years, from communities in Arizona, southwestern Oklahoma, and South and North Dakota participated in the first SHS examination, conducted from 1989 to 1991 (phase 1). The cohort was followed and re-examined twice and has been under continuous yearly surveillance for CV events. The second examination evaluated 89% of all surviving members of the original cohort, who also underwent standard Doppler echocardiography. Thus, the second SHS examination was used as baseline for the present analysis.

From the population of 2794 participants available for the analysis, we excluded 441 participants with prevalent CV (220 with a history of myocardial infarction or cardiac ischemic disease, 44 with a history of stroke and 177 with chronic heart failure) and 94 for low ejection fraction (i.e <50%), 14 because of serum triglycerides (>750 mg/dL), and 333 participants because of incomplete echocardiographic assessment. Thus, for the present study, we included 1912 SHS participants (age 59 ± 8 years; 64% women) with baseline preserved EF and without prevalent CV disease. Institutional review boards of the participating institutions and the participating tribes approved the study and submission of the manuscript.

### 2.2. Measurements and Definitions

The SHS used standard methodology and strict quality control at each clinical examination [11], which included a personal interview, physical examination with anthropometric and blood pressure measurement, and morning blood sample collection after a 12-hour fast. Exams were performed at local community settings and Indian Health Service clinics by trained study staff.

Arterial hypertension was defined as blood pressure ≥140/90 mmHg or current antihypertensive treatment. Obesity was classified as body mass index ≥30 kg/m^2^. Diabetes was defined as fasting glucose ≥126 mg/dL or use of antidiabetic medication. Hyperlipemia was defined as total cholesterol >200 mg/dL and/or triglyceridemic value >150 mg/dL. Smoking habits were defined as non-smoker, former smoker, and current smoker.

### 2.3. Echocardiography

Echocardiograms were performed using phased-array machines, with M-mode, two-dimensional, and Doppler capabilities, as previously reported [12]. Echocardiograms were evaluated in the Core Laboratory at the Weill Cornell Medical College in New York by expert readers blinded to the participant’s clinical details, using a computerized review station (Digisonics, Inc., Houston, TX, USA) equipped with digitizing tablets and monitor screen overlays for calibration and performance of each needed measurement. Reproducibility of echocardiographic measures was tested in the Weill Cornell adult echocardiography laboratory in an ad hoc designed study [13].

LV internal dimensions and wall thickness were measured as end-diastole and end-systole, respectively, as previously reported [12]. Relative wall thickness, LV mass, and LV mass index (by normalization for height in m^2.7^) were also estimated [6]. LVH was defined with LV mass index >47 g/m^2.7^ for both sexes, a validated population-specific cut-point, maximizing the population risk attributable to LVH [14]. SV was calculated as the difference between LV end-diastolic and end-systolic volumes by the z-derived method [6,15]. EF was obtained by the ratio of SV to end-diastolic volume. Midwall fractional shortening was measured as previously described [16].

The ratio of early to late peak diastolic velocities (E/A ratio) was measured as previously described [17]. Based on previous analyses in the SHS, the E/A ratio was categorized as “normal” when it was between 0.6 and 1.5, in “prolonged relaxation” when it was <0.6 and in “restrictive physiology” when it was >1.5 [17].

To assess MEE, we estimated SW as the product of systolic BP times SV (mmHg × mL). Myocardial oxygen consumption (MVO2) could be estimated using the “double product” (DP) of systolic BP × heart rate [18]. Using this second method, MEE may be estimated in mL/s:(1)$$MEE=\frac{SW}{DP}\approx \frac{mmHg\times mL}{mmHg\times bpm}=\frac{mL}{bpm\times 60^{-1}}=\frac{mL}{s}$$

Because of the reportedly close dependence of MEE on LV mass, normalization for LV mass was done to estimate energetic expenditure per unit of myocardial mass (MEEi in mL/s/g) [9].

### 2.4. Outcome

CV events were recorded and adjudicated as previously reported using standardized criteria. [10]. The end-point of the present study was the first occurrence of HF, defined by the Framingham criteria for HF, as previously described [7]. Due to the potential interference with the analyzed outcome, occurrence of acute myocardial infarction (AMI) prior to HF was also censored for analysis as a competing risk event.

### 2.5. Statistical Analysis

Data were analyzed using IBM-SPSS-statistics (version 23.0; SPSS, New Jersey), and expressed as mean ± 1SD. MEEi was categorized in quartiles and analyzed in exploratory analyses, using linear contrast for trend analysis for age and Kendall’s tau as a test for monotonic trends with categorical variables. Cumulative incidences of HF in quartiles of MEEi were analyzed using a Kaplan-Meier plot. Further analyses were focused on the lowest MEEi (corresponding to 25th percentile of the distribution).

We calculated hazard ratios and 95% confidence intervals (CI) of incident HF, using three sequential models of Cox regression. In the first Cox model, the outcome was analyzed in relation to LVH and patterns of E/A ratio, adjusting for age and sex. In the second model, low MEEi, hypertension and anti-hypertensive therapy (no/yes) were forced into the model. In the third model, obesity and diabetes were added to explore how the previous models could be changed by the co-presence of additional CV risk factors. Since the end-point of the present analysis, HF, can also be a consequence of a preceding AMI, a Cox regression was run using AMI preceding HF as a competing risk event. Thus, we censored AMI occurring before HF, in competition with the primary predictor, MEEi [19].

## 3. Results

Among 1912 SHS participants with normal EF and without prevalent CV disease included in this analysis, prevalence of arterial hypertension, obesity, and diabetes were 27, 51, and 40%, respectively.

Table 1 shows that while age was similar among quartiles of MEEi, the proportion of women was progressively lower with decreasing quartiles of MEEi, and concentric LV geometry and LVH were progressively higher (all *p* for trend <0.0001), paralleling the progressive increase in prevalent hypertension, obesity, and diabetes (all *p* for trend <0.0001). There was a significant trend for mitral E/A ratios of <0.6 to sharply increase in the lowest quartile of MEEi, whereas mitral E/A ratios of >1.5 progressively decreased with decreasing quartiles of MEEi (all *p* for trend <0.0001). Hyperlipemia and smoking habits were not different within quartiles of MEEi.

Although only patients with initially normal EF were included in this study, 12% of variability of EF was explained by MEEi, but the explained variability rises to 42% when LV systolic function was evaluated by midwall shortening.

During follow-up studies (median 9.9 years, inter-quartile range 9.3–10.4 years), 126 (7%) participants developed HF, 94 of them after AMI. A Kaplan-Meier cumulative hazard plot demonstrated a significant log-rank, substantially due to the marked deviation of the lowest quartiles of MEEi (Figure 1).

As seen in Figure 1, we compared the lowest MEEi quartile (i.e. ≤0.34 mL × s^−1^ × g^−1^) with all others, defined for convenience as “normal MEEi”. Low MEEi was present in 47% of the subgroup, compared to 24% in the subgroup without incident HF (*p* < 0.0001). Table 2 shows sequential models of Cox regressions for incident HF. Low MEEi predicted incident HF after adjustment for LVH and prolonged relaxation. The impact of low MEEi was reduced after the inclusion of diabetes and smoking habits into the model.

The Cox models were also run using continuous variables for systolic blood pressure, body mass index, MEEi, and LV mass index instead of categories, without modifications, and compared to what has been reported in Table 2. Specifically, for each unit of increasing MEEi, there was a significant 2% reduction of hazard of incident-adverse CV events (hazard ratio (HR)= 0.02; 95% CI 0.002–0.347; *p* < 0.006).

A multicollinearity test was performed using all covariates of model 3 to calculate the variance inflation factor (VIF). The value of VIF was always <1.9, demonstrating optimal performance of the model and the low level of multicollinearity between LVH and low MEEi.

## 4. Discussion

Our analysis demonstrates that in a population-based study with initially normal left ventricular ejection fraction (LVEF), reduced myocardial mechano-energetic efficiency for each g of myocardial mass is a strong predictor of incident HF after adjustment for LVH, prolonged relaxation, and associated CV risk factors, including hypertension, obesity, diabetes, and smoking habits. Our analysis merged CV risk factors with markers of preclinical CV disease. The causative effect of primary risk factors was largely offset by their direct effects on the CV system. The only risk factor that could not be fully offset by CV phenotype was diabetes, which in fact remains a potent risk factor for HF, even after adjustment for CV phenotype, as we have previously demonstrated [20].

In our analysis, we provided an estimation of myocardial energetic efficiency using a very simple method on the basis of a simple assumption, which has been already used in different circumstances [5,18], i.e., that MVO2 consumption mainly depends on developed pressure and frequency of contraction. More complex models of estimating myocardial oxygen consumption have been proposed, with strong rationale, but are likely less suitable for a clinical use [21].

### 4.1. The Conundrum of Development of HF

It is not surprising that a condition of low myocardial mechano-energetic efficiency per unit of myocardial mass significantly contributes to identifying a CV phenotype at risk of developing HF. The link between alterations of myocardial energy balance and HF should merit more attention, especially in the setting of HFpEF. Although we do not have follow-up echocardiograms to compare, we may postulate that incident HFpEF is frequent in our population sample, because our hazard analysis was controlled for incident, intercurrent AMI as a competing risk factor for incident HF, indirectly minimizing the chance that post-ischemic systolic HF could play a substantial role in our findings. Given the frustrating results related to the attempt to improve outcomes in HFpEF [22] and related to the insufficient understanding of its mechanisms, shifting the attention from hemodynamics and cardiac mechanics to the process of production and utilization of energy might be productive [23].

Although increased LV mass is in fact a critical marker of risk, considerable heterogeneity can be found, especially in the setting of HFpEF. In clinical trials and contemporary registries, approximately one-third to two-thirds of patients with HFpEF do not exhibit clear-cut LVH [24]. A proportion of HFpEF patients exhibit eccentric LVH rather than the more usual concentric pattern [24]. Even more intriguing is the evidence that approximately 50% of patients with HFpEF and normal LV mass do not have hypertension [24].

Furthermore, it is noteworthy that in this population with normal EF at baseline, MEEi could explain as much as 42% of midwall shortening variability, compared to the expected negligible correlation with ejection fraction. This finding is physiologically consistent with the assumption that LV systolic chamber function is only a very rough indicator of the status of myocardial mechanics.

Other pathogenetic mechanisms should be investigated.

### 4.2. Diastolic Dysfunction 

Diastolic dysfunction is considered especially important in the context of HFpEF [24]. However, in echocardiographic sub-studies, one-third of patients randomized in controlled trials of HFpEF exhibited normal diastolic function, and a further 20% to 30% only had mild or grade 1 diastolic dysfunction [24]. In addition, a recent study of elderly subjects (age 67 to 90 years) without HF found that 96% of them had abnormal diastolic function, according to guideline-based definitions. In a condition in which diastolic dysfunction has been considered a definite pathophysiologic feature (“diastolic heart failure”), it is unclear whether the absence of diastolic dysfunction in HFpEF reflects a limitation of echocardiography, at least at the light of the present recommendations, or suggests pathophysiological mechanisms that are independent of diastolic function in a substantial proportion of patients. But, perhaps even more importantly, HF is always characterized by increased filling pressure (and therefore real diastolic dysfunction), no matter whether or not EF is reduced.

### 4.3. LV Production, Delivery, and Utilization of Energy

The evidence of impaired myocardial energy balance adds pathophysiological rationale to the strong effect of increased LV mass function as a marker of risk for HF. The index presented in this and in previous longitudinal analyses [9] allows a concentration of attention on physiological mechanisms more related to the production, delivery, and utilization of energy. While normal hearts mainly oxidize fatty acids to produce energy (ATP), hearts with stage B HF require a shift of production of energy (ATP) toward the most convenient glucose-pyruvate oxidation, a shift that implies adequate insulin sensitivity [25].

An important possible mechanism reducing myocardial efficiency is in fact insulin resistance [26]. The emerging evidence that this index is influenced by conditions of insulin resistance [25] is an indirect validation of our physiologic postulate. This is evident in the present and previous analyses in which diabetes exhibits a substantial importance as a predictor of HF, even more than hypertension (confirmed in the present analysis, as seen in Table 2). The progressive decline of MEEi with the increasing prevalence of diabetes and obesity was demonstrated previously in our Italian registry of hypertensive patients [9,25]. Insulin resistance, typical of type 2 diabetes, results in difficulty in the utilization of glucose [27], increasing reliance on fatty acid oxidation for up to 80–90% of acetyl CoA, at the expense of glucose and lactate oxidation [28]. This increase in fatty acid oxidation is the main determinant of the increased MVO2 at zero work, because of the lower oxygen efficiency for ATP synthesis using fatty acids as substrates [29].

Due to the above considerations and given the ethnic specificity of the SHS, and the particular high prevalence of diabetes and obesity, our findings are not necessarily generalizable and might need to be clarified in other populations with different genetic and environmental backgrounds, especially because algorithms for risk prediction might be substantially affected by prevalence and distribution of individual risk factors [30].

### 4.4. Final Considerations

Despite the strong association with incident HF, many unconsidered factors could have an impact on the progression of HF, potentially reducing the impact of baseline low myocardial mechano-energetic efficiency during follow-up. Among them, the control of diabetes or blood pressure during follow-up could have a significant impact on the progression of diastolic dysfunction and precipitation of HF. Further studies should help clarifying pathophysiological mechanisms linking myocardial mechano-energetic efficiency to diastolic dysfunction and control of blood pressure and diabetes.

## 5. Conclusions

This study demonstrated that depressed LV mechano-energetic efficiency per unit of LV mass, computed by using a simple approach comparing external work with the estimated oxygen consumption, is a powerful predictor of incident HF after adjustment for LVH and other confounders in an unselected population-based cohort of American Indians with normal baseline EF. Our results might serve as a hypothesis when testing for a better evaluation of pathophysiology of HFpEF.

## Figures and Tables

**Figure 1 jcm-08-01044-f001:**
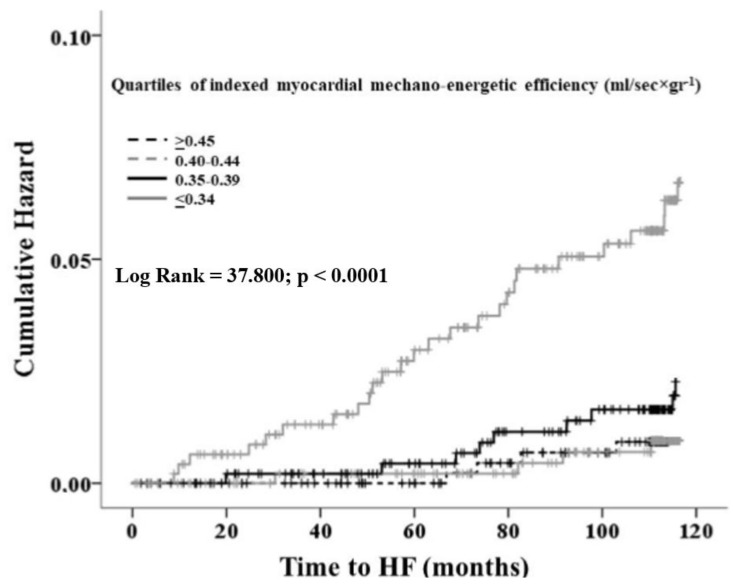
Cumulative hazard of incident heart failure (HF) for quartiles of myocardial mechano-energetic efficiency per unit of left ventricular mass (MEEi). Continuous grey line represents the lowest MEEi quartile.

**Table 1 jcm-08-01044-t001:** Characteristics of quartiles of LV mass-normalized myocardial mechano-energetic efficiency (MEEi).

		Quartiles of Indexed Myocardial Mechano-Energetic Efficiency			
	Whole Population	≥0.45	0.40–0.44	0.35–0.39	≤0.34
	(*n* = 1912)	(*n* = 478)	(*n* = 477)	(*n* = 479)	(*n* = 478)
Age (years)	59 ± 8	59 ± 8	60 ± 8	59 ± 8	60 ± 8
Hypertension (%) ^a^	27%	22%	25%	29%	34%
Proportion of women (%) ^a^	64%	68%	69%	65%	55%
Concentric LV geometry (%) ^a^	4%	0.2 %	1%	2%	11%
LV Hypertrophy (%) ^a^	23%	9%	18%	23%	40%
Mitral E/A ratio <0.6 (%) ^a^	4.1	2.1	2.5	1.3	10.5
Mitral E/A ratio >1.5 (%) ^a^	2.6	4.5	3.4	1.5	1.1
Obesity (%) ^a^	51%	40%	51%	57%	58%
Diabetes (%) ^a^	40%	25%	37%	41%	57%
Hyperlipemia (%)	58	57	55	59	62
Former smoker (%)	35	33	34	36	38
Current smoker	36	39	35	34	35

LV = left ventricular; ^a^ Kendall’s τ-b: all *p* < 0.0001.

**Table 2 jcm-08-01044-t002:** Sequential models of proportional hazard analysis of incident heart failure (HF) in relation to low MEEi.

Predictors	Model 1			Model 2			Model 3		
	*p*	HR	95%CI	*p*	HR	95%CI	*p*	HR	95%CI
Age (years)	0.004	1.04	1.01–1.06	0.007	1.04	1.01–1.06	0.001	1.05	1.02–1.08
Female sex	0.666	0.93	0.62–1.38	0.846	0.96	0.63–1.46	0.833	1.05	0.68–1.61
LV Hypertrophy	<0.0001	2.51	1.70–3.73	0.001	2.01	1.37–3.10	0.004	1.89	1.23–2.91
E/A <0.6	<0.0001	3.72	1.99–6.98	0.002	2.85	1.48–5.51	0.004	2.60	1.35–5.05
E/A >1.5	0.612	0.60	0.08–4.33	0.629	0.61	0.09–4.43	0.800	0.77	0.11–5.60
Low MEEi				0.005	1.83	1.21–2.79	0.026	1.61	1.06–2.44
Hypertension				0.484	1.27	0.66–2.45	0.672	1.15	0.60–2.23
Anti-hypertensive therapy (y/n)				0.012	2.28	1.20–4.35	0.094	1.75	0.91–3.35
Diabetes							<0.0001	3.11	2.01–4.80
Obesity							0.191	0.76	0.50–1.15
Hyperlipemia							0.832	0.96	0.64–1.43
Former smoker							0.006	2.11	1.24–3.60
Current Smoker							0.003	2.38	1.35–4.17

LV = left ventricular; MEEi = indexed myocardial mechano-energetic efficiency; HR = hazard ratio.

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
