# Peer review of "Depressed Myocardial Energetic Efficiency Increases Risk of Incident Heart Failure: The Strong Heart Study"

_jcm, 2019, doi:10.3390/jcm8071044_

Round 1
Reviewer 1 Report
A brief summary
Authors showed that myocardial mechano-energy efficiency (MEE) and MEE index, defined as MEE/LV mass reflects LV oxygen consumption is a predictor of incident heart failure in patients with hypertensiton with normal LV ejection fraction.
Broad comments:
These idea and design are clear and serve ideas to evaluate of pathophysiology of HFpEF as authors said. However, authors have already shown that MEEi was a predictor of cardiovascular events including HF in patients with hypertension and normal EF in the Journal of Hypertension 2016.
Major comments
1. Authors have already shown that MEEi was a predictor of cardiovascular events including HF in patients with hypertension and normal EF in the Journal of Hypertension 2016. The study cohort are different, but we thought that these results and the result of the present study are similar. So new information and study impact of the present study is low.
2. Authors said that 12% of variability of EF was explained by MEEi, but the explained variability rises to 42% when LV systolic function was evaluated by midwall shortening. You should show raw data or figure of this redults.
3. Mitral E/A ratio <0.6 was included in Cox proportional models. If Mitral E/A ratio >1.5 include in Cox models, would be the results similar?
4. You showed that contents of antihypertensive treatments affected both MEEi and cardiovascular events in the past study. In the present study, do specific antihypertensive treatments affect Meei and/or incidence of HF?
Author Response
To Reviewer # 1
We appreciate your thoughtful comments and valuable suggestions to improve impact of our paper.. In this letter we used red for our answers, and black for your comments.
We addressed your comments as follows:
Authors showed that myocardial mechano-energy efficiency (MEE) and MEE index, defined as MEE/LV mass reflects LV oxygen consumption is a predictor of incident heart failure in patients with hypertension with normal LV ejection fraction.
In our previous paper, we used as a composite end-point, including a few cases of heart failure (we have relatively low prevalence of HF in our registry). Of note, that paper refers to a registry of treated hypertensive patients, whereas the SHS is a population-based cohort of American Indians and in the specific field of HF development. We do not believe the two studies are comparable.
Broad comments:
These idea and design are clear and serve ideas to evaluate of pathophysiology of HFpEF as authors said. However, authors have already shown that MEEi was a predictor of cardiovascular events including HF in patients with hypertension and normal EF in the Journal of Hypertension 2016.
As we pointed out before, the contest in which we tested our hypothesis was completely different, in the type of population and the end-point, as well. Also, as pointed out in the discussion, the SHS is an ethnic cluster, a characteristic that has been remarked in all analyses, together with the warning that automatic extension to other ethnicities need prudence.
Major comments
1. Authors have already shown that MEEi was a predictor of cardiovascular events including HF in patients with hypertension and normal EF in the Journal of Hypertension 2016. The study cohort are different, but we thought that these results and the result of the present study are similar. So new information and study impact of the present study is low.
We think that we clearly expressed our point of view.
2. Authors said that 12% of variability of EF was explained by MEEi, but the explained variability rises to 42% when LV systolic function was evaluated by midwall shortening. You should show raw data or figure of this results.
Below, you find the scatterplot of MEEi vs EF or midwall shortening. We do not think this figure is critical, for the clarity of our paper and our goals, but if you and the editor think differently, we will be happy to add it to the manuscript.
3. Mitral E/A ratio <0.6 was included in Cox proportional models. If Mitral E/A ratio >1.5 include in Cox models, would be the results similar?
We thank the reviewer for this comment. In fact, we used the classification proposed by Bella et al (ref # 17), by classifying diastolic function in abnormal relaxation (i.e E/A <0.6) normal (i.e. E/A between 0.6 and 1.5) and pseudo-normal (i.e. E/A >1.5) pattern, as already reported in the methods section; the omission from the table was just an error that we have fixed. Please note that the table has been slightly modified according to a 2nd Reviewer’s request (addition of hypercholesterolemia and smoking habit).
4. You showed that contents of antihypertensive treatments affected both MEEi and cardiovascular events in the past study. In the present study, do specific antihypertensive treatments affect Meei and/or incidence of HF?
The design of the SHS includes control check every 4-5 years. It is difficult to dissect single class of meds effects. However, we do have data on antihypertensive therapy in the form no/yes that are reliable and may be used in this type of analysis. We did it in our multivariate Cox model (Table 2)

Reviewer 2 Report
This is a manuscript about the predictive value of myocardial mechano-energetic efficiency for the incident of non-AMI related HF in subjects with initial normal EF. This is a fine study, however there are some issues to be addressed. In particular, it was amazing that baseline MEEi significantly affected the long term risk of HF. There might be many factors that might have more impact on the progression of HF than baseline MEEi during long time follow-up. For instance, the control of diabetes or blood pressure during follow-up might have significant impacts on the progression of diastolic dysfunction. The explanation about it should be added.
#1 Was there any predictive value of MEEi for the incidence of AMI? If so, what is the mechanism of it? How about the association between MEEi and coronary risk factors? In addition the data about dyslipidemia and smoke should be added.
#2 How about the data of medications? Some medications significantly affect the value of MEEi and the risk of HF occurrence.
Author Response
To Reviewer # 2
We appreciate your thoughtful comments and valuable suggestions to improve impact of our paper. In this letter we used red for our answers, and black for your comments.
We addressed your comments as follows:
This is a manuscript about the predictive value of myocardial mechano-energetic efficiency for the incident of non-AMI related HF in subjects with initial normal EF. This is a fine study, however there are some issues to be addressed. In particular, it was amazing that baseline MEEi significantly affected the long-term risk of HF. There might be many factors that might have more impact on the progression of HF than baseline MEEi during long time follow-up. For instance, the control of diabetes or blood pressure during follow-up might have significant impacts on the progression of diastolic dysfunction. The explanation about it should be added.
We agree. A paragraph has been added at the end of Discussion (page XX, lines XX-XX):
“Final considerations.
Despite the strong association with incident HF, many unconsidered factors could have impact on the progression of HF potentially reducing the impact of baseline low myocardial mechano-energetic efficiency during follow-up. Among them, the control of diabetes or blood pressure during follow-up could have significant impact on the progression of diastolic dysfunction and precipitation of HF. Further studies should help clarifying pathophysiological mechanisms linking myocardial mechano-energetic efficiency to diastolic dysfunction and control of blood pressure and diabetes.”
#1 Was there any predictive value of MEEi for the incidence of AMI? If so, what is the mechanism of it? How about the association between MEEi and coronary risk factors? In addition, the data about dyslipidemia and smoke should be added.
There was not a significant predictive value of MEEi for incident AMI [HR2.99 (95% Cl 0.04-2.06); p=0.220] ). However, as pointed out in the methods section, subheading Statistics: “Since the end-point of the present analysis, HF, can also be a consequence of a preceding AMI, Cox regression was run using AMI preceding HF, as a competing risk event.”
The reason we did not use hypercholesterolemia as covariate is in the low predictive value of circulating cholesterol level in the presence of obesity (see Miettinen TA, Circulation, 1971; XLIV:842). According to your request, however, the Cox model is now including also hyperlipidemia and smoking habit.
We used a variable hyperlipidemia where patients were dichotomized basing on total cholesterol (abnormal if >200mg/dL) and/or on triglyceridemic value (abnormal if >150
mg/dL), whereas 3 subgroups of smoking habit were considered: no smoker, previous smoker, current smoker. Percentage of patients with hyperlipemia and smokers were also reported in table 1. You can see that no differences were found among quartiles of MEEi.
In addition, we added these variables in the Cox model. Results did not change concerning the impact of MEEi, but smoking habit became significant, as you can check in Table 2.
#2 How about the data of medications? Some medications significantly affect the value of MEEi and the risk of HF occurrence.
The design of the SHS includes control check every 4-5 years. It is difficult to dissect single class of meds effects. However, we do have data on antihypertensive therapy in the form no/yes that are reliable and may be used in this type of analysis. We did it However in our multivariate Cox model (Table 2).

Round 2
Reviewer 1 Report
I do not have additive comments. Authors responded appropriately.
Reviewer 2 Report
The revised manuscript was finely corrected.